# Comorbidity and clinical factors associated with COVID-19 critical illness and mortality at a large public hospital in New York City in the early phase of the pandemic (March-April 2020)

**Thomas D. Filardo**[1]*, **Maria R. Khan**[2], **Noa Krawczyk**[2], **Hayley Galitzer**[3], **Savannah Karmen-Tuohy**[3], **Megan Coffee**[1,4], **Verity E. Schaye**[4,5], **Benjamin J. Eckhardt**[1,4], **Gabriel M. Cohen**[1,4]

1 Division of Infectious Diseases and Immunology, New York University Grossman School of Medicine, New York, NY, United States of America, 2 Department of Population Health, New York University Grossman School of Medicine, New York, NY, United States of America, 3 New York University Grossman School of Medicine, New York, NY, United States of America, 4 NYC Health + Hospitals, Bellevue Hospital Center, New York, NY, United States of America, 5 Department of Medicine, New York University Grossman School of Medicine, New York, NY, United States of America

* Thomas.Filardo@nyulangone.org

## Abstract

### Background

Despite evidence of socio-demographic disparities in outcomes of COVID-19, little is known about characteristics and clinical outcomes of patients admitted to public hospitals during the COVID-19 outbreak.

### Objective

To assess demographics, comorbid conditions, and clinical factors associated with critical illness and mortality among patients diagnosed with COVID-19 at a public hospital in New York City (NYC) during the first month of the COVID-19 outbreak.

### Design

Retrospective chart review of patients diagnosed with COVID-19 admitted to NYC Health + Hospitals / Bellevue Hospital from March 9th to April 8th, 2020.

### Results

A total of 337 patients were diagnosed with COVID-19 during the study period. Primary analyses were conducted among those requiring supplemental oxygen (n = 270); half of these patients (135) were admitted to the intensive care unit (ICU). A majority were male (67.4%) and the median age was 58 years. Approximately one-third (32.6%) of hypoxic patients managed outside the ICU required non-rebreather or non-invasive ventilation. Requirement

**Data Availability Statement:** All relevant data are within the manuscript and its Supporting Information files.

**Funding:** The authors received no specific funding for this work.

**Competing interests:** The authors have declared that no competing interests exist.

of renal replacement therapy occurred in 42.3% of ICU patients without baseline end-stage renal disease. Overall, 30-day mortality among hypoxic patients was 28.9% (53.3% in the ICU, 4.4% outside the ICU). In adjusted analyses, risk factors associated with mortality included dementia (adjusted risk ratio (aRR) 2.11 95%CI 1.50–2.96), age 65 or older (aRR 1.97, 95%CI 1.31–2.95), obesity (aRR 1.37, 95%CI 1.07–1.74), and male sex (aRR 1.32, 95%CI 1.04–1.70).

## Conclusion

COVID-19 demonstrated severe morbidity and mortality in critically ill patients. Modifications in care delivery outside the ICU allowed the hospital to effectively care for a surge of critically ill and severely hypoxic patients.

## Introduction

COVID-19, the disease caused by infection with severe acute respiratory syndrome coronavirus 2 (SARS-CoV-2) was first identified in China in late 2019 and subsequently spread globally. The WHO declared the outbreak of COVID-19 a pandemic on March 11, 2020. [1] In March of 2020, New York City (NYC) became the epicenter of the outbreak in the United States, experiencing over 219,030 cases, 18,800 confirmed deaths, and 4,624 probable deaths as of July 21, 2020 [2, 3]. During the peak of the COVID-19 surge, NYC hospitals were forced to adapt in order to meet the needs of an overwhelming number of cases; bed capacity was expanded and clinical triage and treatment algorithms were rapidly implemented.

COVID-19 presents with a broad spectrum of illness, ranging from asymptomatic or mild infections to severe disease leading to critical illness and multi-organ failure. Advanced age and comorbid medical conditions, such as hypertension and diabetes, have been shown to predispose individuals to severe illness due to COVID-19 [4–8]. Prior studies have shown that patients admitted to the intensive care unit (ICU) due to COVID-19 have high mortality [9–11]. Furthermore, critical illness due to COVID-19 can be prolonged, straining hospital systems beyond their typical capacity [10–12].

In this study we describe patients admitted to NYC Health + Hospitals / Bellevue Hospital Center (BHC), a large, academic, public safety-net hospital in NYC, with COVID-19 in the month following the hospital's first diagnosed case. In particular, we focus on comorbidities, clinical characteristics, and outcomes for patients admitted to ICU and non-ICU settings.

## Methods

### Study setting

BHC is an 844-bed quaternary-care referral center for the NYC Health + Hospitals (NYCH +H) system. NYCH+H comprises 11 acute care hospitals and is the public safety-net healthcare system for NYC. NYCH+H facilities were on the front lines of the response to the COVID-19 pandemic, serving vulnerable populations throughout the city [13]. This study was approved by the New York University Grossman School of Medicine and NYCH+H Institutional Review Boards.

## Data

This is a retrospective review of the electronic medical record (EMR) of patients aged 18 and older admitted to BHC with laboratory-confirmed COVID-19 between March 9th 2020 and April 8th 2020. Laboratory-confirmed COVID-19 was defined as having a positive reverse transcriptase polymerase chain reaction (RT-PCR) test for SARS-CoV-2 virus. Patients were tested using a variety of diagnostic tests, including those performed by the NYC Department of Health and Mental Hygiene (DOHMH) (started 2/5/2020), as well as Labcorp (started 3/8/2020) and Bioreference (started 3/19/2020) commercial laboratories. BHC began using an in-house test (Cepheid GeneXpert Xpress) on 4/1/2020. Patients diagnosed at affiliated NYCH +H hospitals and transferred to BHC during the study period were included in this analysis, with data available via an integrated EMR. Incarcerated individuals were excluded as a vulnerable population.

Retrospective manual data extraction was conducted using a standardized data capture tool. Data was gathered between April 8, 2020 and May 10, 2020. Data extracted from the EMR were entered into a secure REDCap (research electronic data capture) database [14].

## Measures

Data included demographics, comorbid conditions, presenting symptoms, admission vital signs and laboratory values, level of respiratory support, treatment, complications, and hospital course (i.e. ICU transfers, discharge, death). Obesity was defined as a body mass index of 30.0 or greater. Laboratory values on hospital days 0, 1, and 2 were included as admission labs. Only the first available lab value was included if multiple were taken during this timeframe. If a patient was diagnosed with COVID-19 7 or more days after the date of hospital admission, the date of COVID-19 testing was considered hospital day 0 for this analysis. Otherwise, the day of admission was considered hospital day 0. Acute kidney injury (AKI) was defined as an increase in serum creatinine 0.3mg/dL above baseline in 48 hours or less. Use of therapies targeting SARS-CoV-2 and acute respiratory distress syndrome (ARDS) were gathered.

Race and ethnicity were gathered from patient-reported data entered into the EMR. Options available for race in the EMR included American Indian or Alaskan, Asian, Black, Native Hawaiian, Other Pacific Islander, White, Other, and Unknown/Not Recorded. Ethnicity included assessment of Hispanic / Latinx or Non-Hispanic identity. All patients who reported Hispanic / Latinx ethnicity reported "Other" or "Unknown/Not Recorded" race and were categorized as "Hispanic / Latinx" in a condensed assessment of race and ethnicity.

Our primary outcome was mortality at 30 days. Secondary outcomes included discharge from the hospital by 30 days, persistent hospitalization at 30 days, use of mechanical ventilation and other respiratory support, and incidence of AKI with or without requirement of renal replacement therapy (RRT). Patients with baseline end-stage renal disease (ESRD) were excluded from outcomes regarding renal injury.

Patients were assessed by their highest level of oxygen and ventilatory support, in ascending order of supplementation: no supplemental oxygen, oxygen via nasal cannula or simple facemask at 6L/min or less, oxygen via non-rebreather (NRB), respiratory support via non-invasive ventilation or high-flow nasal cannula (NIV/HFNC), or mechanical ventilation.

The National Early Warning Score (NEWS) was calculated for patients based on admission vitals (temperature, heart rate, respiratory rate, systolic blood pressure, pulse oximetry) and altered mental status on presentation was assessed based on provider documentation [15].

## Data analysis

Socio-demographics, clinical characteristics, and laboratory markers at admission were compared across patients with COVID-19 illness who did and did not require admission to the ICU. Given a non-normal distribution of continuous variables, as determined by a Shapiro-Wilk normality test, we presented the median and interquartile range for continuous variables and compared continuous variables using Wilcoxon rank-sum tests. For categorical variables, we reported count and proportion for each group, and used Fisher's exact test to compare distributions across categorical variables due to the small size of the sample.

Using binomial regression, we estimated unadjusted and adjusted risk ratios (RRs) and 95% confidence intervals (CIs) for associations between socio-demographic and comorbidity factors and mortality due to COVID-19 among patients requiring supplemental oxygen. Multivariable models included adjustment for all socio-demographic and comorbidity factors in order to estimate independent associations between independent variables and mortality. We also estimated RRs and 95% CIs for associations between laboratory indicators and mortality. Because laboratory indicators may offer improved clinical utility for specific subgroups based on demographics and/or presence of preexisting conditions, we tested for effect modification of the association between each laboratory indicator and mortality by age, sex, and presence of any comorbidity (cardiovascular, pulmonary, or renal comorbidity, diabetes, immunosuppression, HIV, malignancy, dementia, and obesity). In cases where associations differed by background factors (age, sex, presence of comorbidity), as indicated by significance of the background factor by laboratory indication product interaction term (P, 0.15), subgroup associations were presented. Otherwise, associations were presented in the overall sample to preserve statistical power and included adjustment for age, sex, and presence of any comorbidity. All models estimating associations between laboratory factors and mortality were adjusted for race/ethnicity; however, we did not assess effect modification by race/ethnicity because low cell counts in some race categories would not have supported reliable estimation of associations.

All statistical tests were considered to be significant at the $p<0.05$ level, with the exception of the interactions tests that were considered significant at the $p<0.15$ level. The significance level for interaction analyses was increased to avoid type II error [16]. Data analysis was completed in Stata, version 15.0 (Stata Corp., College Station, TX).

## Results

### Study population and demographics

Between March 9th and April 8th, 2020, a total of 337 adult patients eligible for inclusion in this study were diagnosed with COVID-19 at BHC. This cohort was split first by need for supplemental oxygen, with 65 patients (19.3%) never requiring supplemental oxygen. This group represented a heterogeneous population, including many admitted from congregate settings early in the outbreak for social isolation and individuals already hospitalized for other reasons (e.g. inpatient psychiatric care, acute inpatient rehab). Details regarding the 65 patients never requiring supplemental oxygen are available in S1 Table. Of the remaining 272 patients requiring supplemental oxygen, 2 patients were transferred to outside hospitals prior to hospital day 30, and therefore 30-day hospital outcomes were unable to be assessed. Primary analyses were conducted among the remaining patients who required supplemental oxygen (n = 270).

These remaining 270 patients requiring supplemental oxygen were divided by requirement of ICU admission, with 135 (50.0%) requiring ICU admission during the study period. The median age was 58, with 62 (23.0%) patients under age 50. Approximately two-thirds (67.4%)

were male. Detailed information regarding race and ethnicity was available for 126 (46.7%) participants, with a minority identifying as white (9.3%). A total of 15 patients (5.6%) were homeless, either street- or shelter-domiciled (Table 1).

## Baseline comorbid medical conditions

The majority of patients had at least one comorbid medical condition, with cardiovascular conditions (51.5%), obesity (41.8%), and type 2 diabetes (33.0%) being most prevalent (Table 1). Few patients had baseline pulmonary (13.3%) or renal (11.9%) disease. A total of 50 patients (18.5%) had none of these specifically assessed comorbidities. In general, presence of baseline comorbid conditions did not differ by ICU admission except for chronic kidney disease (CKD), which was more prevalent in the ICU population (11.1% vs 3.0%, p = 0.02). Only 13 patients (4.8%) reported being active smokers (S2 Table).

## Clinical presentation

The most common presenting symptoms were fever (83.0%), cough (80.0%) and dyspnea (77.0%). There were no significant differences in symptom presentation when stratified by ICU admission. Median duration of symptoms prior to presentation was similar in hypoxic patients managed on the floor (7 days) and in the ICU (6 days). Median NEWS score was higher for patients admitted to the ICU (7 vs 6, p = 0.002), and high-risk NEWS scores were more prevalent in patients requiring ICU admission (p = 0.008) (Table 1). Assessment of possible COVID-19 exposures and admission vital signs are included in S2 Table.

Median white blood cell count (8.05 vs 6.49), neutrophil count (6.60 vs 5.25), CRP (172.8 vs 109.4), ferritin (964.3 vs 725.6), and lactate dehydrogenase (LDH) (584 vs 435) were higher in patients admitted to the ICU compared to those not admitted to the ICU (p<0.05) (Table 1). Higher rates of troponin elevation (30.2% vs 13.3%, p = 0.008) and AST elevation (79.0% vs 62.9%, p = 0.005) were also seen in patients admitted to the ICU.

## Treatment

The majority of patients requiring supplemental oxygen received at least one medication targeting SARS-CoV-2 virus (Table 2). The majority of patients received hydroxychloroquine (HCQ; 82.2%), with a higher rate of treatment in patients admitted to the ICU (90.4% vs 74.1%). The majority of patients admitted to the ICU received steroids compared to a minority of patients not admitted to the ICU (71.9% vs 3.0%). Most patients also received antibiotics, with a higher rate of use in patients admitted to the ICU (80.7% vs 53.3%).

Patients managed outside the ICU mostly required nasal cannula alone (67.4%) but 44 of these patients (32.6%) required treatment with NRB or NIV/HFNC for severe hypoxia or respiratory distress. Nearly all patients admitted in the ICU required mechanical ventilation (89.6%) with only 14 (10.4%) requiring either NRB or NIV/HFNC as their highest level of respiratory support (Table 2).

## Renal injury

Of patients requiring ICU admission, 99 of 130 patients without baseline ESRD (76.1%) developed AKI and 55 of these patients (42.3%) required RRT. Of hypoxic floor patients without ESRD (127), 23 had AKI without RRT (18.1%) and 1 patient required RRT (0.8%) (Table 2).

**Table 1. Socio-demographics, medical history, and admission laboratory markers for patients with COVID-19 Illness requiring supplemental oxygen (n = 270).**

| | Never ICU (n = 135) | ICU (n = 135) | Total (n = 270) | p-value[1] |
|---|---|---|---|---|
| **Sex, n (%)** | | | | |
| Male | 87 (64.4%) | 95 (70.4%) | 182 (67.4%) | 0.36 |
| Female | 48 (35.6%) | 40 (29.6%) | 88 (32.6%) | - |
| **Age, median (IQR)** | 57 (48–67) | 60 (51–68) | 58 (50–67) | 0.19 |
| **Age Category, n (%)** | | | | |
| <50 | 36 (26.7%) | 26 (19.3%) | 62 (23.0%) | 0.32 |
| 50–64 | 60 (44.4%) | 63 (46.7%) | 123 (45.6%) | - |
| ≥65 | 39 (28.9%) | 46 (34.1%) | 85 (31.5%) | - |
| **Race/Ethnicity,[2] n (%)** | | | | |
| American Indian or Alaska Native | 0 (0%) | 1 (0.7%) | 1 (0.4%) | 0.42 |
| Asian | 9 (6.7%) | 11 (8.2%) | 20 (7.4%) | - |
| Black | 29 (21.5%) | 27 (20.0%) | 56 (20.7%) | - |
| Hispanic/Latinx | 9 (6.7%) | 15 (11.1%) | 24 (8.9%) | - |
| White | 16 (11.9%) | 9 (6.7%) | 25 (9.3%) | - |
| Other | 61 (45.2%) | 64 (47.4%) | 125 (46.3%) | - |
| Not Recorded or Unknown | 11 (8.2%) | 8 (5.9%) | 19 (7.0%) | - |
| **BMI, median (IQR), n = 261** | 29.02 (25.3–32.4) | 28.86 (25.8–34.2) | 28.96 (25.7–33.2) | 0.20 |
| **Medical Comorbidities, n (%)** | | | | |
| Cardiovascular Comorbidity[3] | 66 (48.9%) | 73 (54.1%) | 139 (51.5%) | 0.47 |
| Pulmonary Comorbidity[4] | 21 (15.6%) | 15 (11.1%) | 36 (13.3%) | 0.37 |
| Renal Comorbidity[5] | 12 (8.9%) | 20 (14.8%) | 32 (11.9%) | 0.19 |
| CKD | 4 (3.0%) | 15 (11.1%) | 19 (7.0%) | 0.02 |
| ESRD | 8 (5.9%) | 5 (3.7%) | 13 (4.8%) | 0.57 |
| Type 2 Diabetes | 42 (31.1%) | 47 (34.8%) | 89 (33.0%) | 0.61 |
| Immunosuppression | 6 (4.4%) | 5 (3.7%) | 11 (4.1%) | 1.00 |
| HIV | 3 (2.2%) | 2 (1.5%) | 5 (1.9%) | 1.00 |
| Malignancy | 6 (4.4%) | 1 (0.7%) | 7 (2.6%) | 0.12 |
| Dementia | 5 (3.7%) | 4 (3.0%) | 9 (3.3%) | 1.00 |
| Obesity (BMI >30.0), n = 261 | 50 (39.1%) | 59 (44.4%) | 109 (41.8%) | 0.45 |
| No Listed Comorbidities[6] | 26 (19.3%) | 24 (17.8%) | 50 (18.5%) | 0.88 |
| **Pregnancy, n (%)** | 2 (1.5%) | 1 (0.7%) | 3 (1.1%) | 1.00 |
| **Homelessness, n (%)** | 8 (5.9%) | 7 (5.2%) | 15 (5.6%) | 1.00 |
| **Symptoms, n (%)** | | | | |
| Fever | 114 (84.4%) | 110 (80.5%) | 224 (83.0%) | 0.63 |
| Cough | 109 (80.7%) | 107 (79.3%) | 216 (80.0%) | 0.88 |
| Dyspnea | 106 (78.5%) | 102 (75.6%) | 208 (77.0%) | 0.67 |
| Chest Pain | 25 (18.5%) | 22 (16.3%) | 47 (17.4%) | 0.75 |
| Diarrhea | 29 (21.5%) | 34 (25.2%) | 63 (23.3%) | 0.57 |
| Myalgias | 45 (33.3%) | 41 (30.4%) | 86 (31.9%) | 0.70 |
| Anosmia | 9 (6.7%) | 5 (3.7%) | 14 (5.2%) | 0.21 |
| Altered Mental Status | 7 (5.2%) | 13 (9.6%) | 20 (7.4%) | 0.25 |
| Headache | 32 (23.7%) | 19 (14.1%) | 51 (18.9%) | 0.06 |
| Syncope | 0 (0%) | 3 (2.2%) | 3 (1.1%) | 0.25 |
| **Symptom Duration[7], median (IQR) (n = 258)** | 7 (4.0–8.5) | 6 (3–8) | 7 (4–8) | 0.75 |
| **NEWS Score, median (IQR), n = 267** | 6 (4–8) | 7 (5–9) | 7 (4–9) | 0.002 |
| **NEWS Score category, n (%), n = 267** | | | | |
| Low Risk (0–4) | 41 (30.6%) | 27 (20.3%) | 68 (25.5%) | 0.008 |

*(Continued)*

**Table 1.** (*Continued*)

| | Never ICU (n = 135) | ICU (n = 135) | Total (n = 270) | p-value[1] |
|---|---|---|---|---|
| Medium Risk (5–6) | 38 (28.4%) | 26 (19.6%) | 64 (24.0%) | - |
| High Risk (≥7) | 55 (41.0%) | 80 (60.2%) | 135 (50.6%) | - |
| **Admission Lab Tests, median (IQR)** | | | | |
| WBC ($10^3$/μl) | 6.49 (5.2–9.3) | 8.05 (6.0–11.0) | 7.5 (5.6–10.3) | 0.003 |
| ANC ($10^3$/μl), n = 269 | 5.25 (3.7–7.8) | 6.60 (4.7–9.6) | 5.97 (4.3–9.1) | 0.002 |
| ALC ($10^3$/μl), n = 269 | 0.99 (0.7–1.2) | 0.82 (0.58–1.11) | 0.91 (0.6–1.2) | 0.07 |
| CRP (mg/L), n = 230 | 109.4 (29.4–182.6) | 172.8 (102.1–228.4) | 147 (69.9–209.2) | <0.001 |
| D-dimer (ng/mL), n = 211 | 258 (192–662) | 430 (311–806) | 379 (222–708) | <0.001 |
| Ferritin (ng/mL), n = 187 | 725.6 (347.5–1259) | 964.3 (575–1382.3) | 867 (465.5–1326.4) | 0.04 |
| LDH (IU/mL), n = 223 | 435 (320–554) | 584 (458.5–773.5) | 505 (372–675) | <0.001 |
| **Admission Lab Tests, n (%)** | | | | |
| Lactate >2.0 (mmol/L), n = 193 | 32 (35.2%) | 33 (32.4%) | 65 (33.7%) | 0.76 |
| Troponin ≥0.05 (ng/mL), n = 189 | 11 (13.3%) | 32 (30.2%) | 43 (22.8%) | 0.008 |
| AST >40 (U/L), n = 265 | 83 (62.9%) | 105 (79.0%) | 188 (70.9%) | 0.005 |
| ALT >36 (U/L), n = 265 | 72 (54.6%) | 81 (60.9%) | 153 (57.7%) | 0.32 |

All variables were calculated for the full sample of 270 patients, unless different (n) indicated in variable column. Median and interquartile range (IQR) presented for continuous variables (age, BMI, symptom duration, NEWS score, and some lab results). Count and proportion presented for remaining categorical variables.

[1]p-values are presented for Wilcoxon rank-sum tests for continuous variables and Fisher's exact tests for categorical variables.

[2]Race/ethnicity categories are presented as recorded in electronic records. See Methods section for details.

[3]Hypertension, Heart Failure, Stroke or Transient Ischemic Attack, Coronary Artery Disease.

[4]Asthma, Chronic Obstructive Pulmonary Disease (COPD), Obstructive Sleep Apnea, Interstitial Lung Disease.

[5]Chronic Kidney Disease (CKD) or End-stage Renal Disease (ESRD).

[6]Patients lacking all comorbid medical conditions included in this table. No patients in this sample had chronic liver disease or cirrhosis, which is omitted.

[7]Patients were omitted if diagnosis occurred >7 days after hospital admission.

## Hospital outcomes

Hospital outcomes stratified by respiratory support and renal injury are available in Table 3. Mortality among those requiring supplemental oxygen was 28.9%: 53.3% (72 of 135) in the ICU and 4.4% (6 of 135) outside the ICU. Half of the patients with fatal outcome outside the ICU had AKI without RRT (3 of 6), and two-thirds required NRB or NIV/HFNC (4 of 6). The majority of deaths occurred in the ICU (72 of 78, 92.3%) and among patients requiring mechanical ventilation (67 of 78, 85.9%). The subset of patients requiring NRB or NIV/HFNC outside the ICU showed a trend towards higher mortality than those requiring nasal cannula alone (9.1% vs 2.2%, p = 0.088). Of ICU patients surviving to discharge or hospital day 30, 45 (33.3%) remained hospitalized with 22 (16.3% of total) still in the ICU; only 18 (13.3% of total) had been discharged home or to a care facility.

In critically ill patients without ESRD at baseline, incidence of AKI without requirement of RRT showed a trend towards higher mortality (52.3% vs 29.0%, p = 0.059) and AKI with RRT was associated with increased mortality (65.5% vs 29.0%, p = 0.002). Among patients outside the ICU without ESRD at baseline, AKI was associated with a trend towards increased mortality (12.5% vs 2.9%, p = 0.081) (Table 3).

In multivariate analysis of risk factors for mortality, controlling for demographics and baseline comorbid conditions, the strongest risk factors for mortality included dementia (aRR 2.11 95%CI 1.50–2.96), age 65 or older (aRR 1.97, 95%CI 1.31–2.95), obesity (aRR 1.37, 95%CI 1.07–1.74), and male sex (aRR 1.32, 95%CI 1.04–1.70). The effect of cardiovascular (aRR 1.12,

**Table 2. Treatments, respiratory support, and renal injury for patients with COVID-19 Illness requiring supplemental oxygen (n = 270).**

| | Never ICU (n = 135) | ICU (n = 135) | Total (n = 270) |
|---|---|---|---|
| **Treatment, n (%)** | | | |
| Lopinavir/Ritonavir | 15 (11.1%) | 13 (9.6%) | 28 (10.4%) |
| Hydroxychloroquine (HCQ) | 25 (18.5%) | 10 (7.4%) | 35 (13.0%) |
| HCQ + Azithromycin | 75 (55.6%) | 112 (83.0%) | 187 (69.3%) |
| Any HCQ | 100 (74.1%) | 122 (90.4%) | 222 (82.2%) |
| Tocilizumab | 3 (2.2%) | 26 (19.3%) | 29 (10.7%) |
| Remdesivir Study Enrollment[1] | 0 (0%) | 4 (3.0%) | 4 (1.5%) |
| Antibiotics | 72 (53.3%) | 109 (80.7%) | 181 (67.0%) |
| Steroids | 4 (3.0%) | 97 (71.9%) | 101 (37.4%) |
| None of the Above | 6 (4.4%) | 2 (1.5%) | 8 (3.0%) |
| **Highest Level of Respiratory Support, n (%)** | | | |
| Oxygen—Nasal Cannula | 91 (67.4%) | 0 (0%) | 91 (33.7%) |
| Oxygen—Non-Rebreather (NRB) | 41 (30.4%) | 5 (3.7%) | 46 (17.0%) |
| NIV/HFNC | 3 (2.2%) | 9 (6.7%) | 12 (4.4%) |
| Mechanical Ventilation | 0 (0%) | 121 (89.6%) | 121 (44.8%) |
| **Renal Injury, n (%)** | | | |
| Baseline ESRD | 8 (5.9%) | 5 (3.7%) | 13 (4.8%) |
| No ESRD | 127 (94.1%) | 130 (96.3%) | 257 (95.2%) |
| No AKI[2] | 103 (81.1%) | 31 (23.8%) | 134 (52.1%) |
| AKI without RRT[2] | 23 (18.1%) | 44 (33.8%) | 67 (26.1%) |
| AKI with RRT[2] | 1 (0.8%) | 55 (42.3%) | 56 (21.8%) |

Data are reported as count and proportion for categorical variables.

[1]Receipt of remdesivir or placebo is unknown for these patients.

[2]Percentages reported from denominator of patients without baseline ESRD.

Abbreviations: NIV Non-invasive Ventilation: HFNC High Flow Nasal Cannula: AKI Acute Kidney Injury: RRT Renal Replacement Therapy: ESRD End-Stage Renal Disease.

95%CI 0.75–1.68) and renal comorbidities (aRR 1.27, 95%CI 0.90–1.78) showed a trend towards higher mortality but did not reach statistical significance. Pulmonary conditions and type 2 diabetes were not associated with mortality (Table 4).

Of laboratory values assessed for association with mortality in adjusted analysis, elevated ferritin (>1000 mg/dL;aRR 1.66 95%CI 1.05–2.63), lactate dehydrogenase (LDH >500 IU/mL; aRR 2.12 95%CI 1.32–3.41), elevated aspartate transaminase (AST >40 U/L; aRR 1.84, 95% CI 1.11–3.04) and troponin (>0.05 ng/mL;aRR 1.57 95%CI 1.03–2.41) were associated with mortality in the overall sample. Neutrophil and lymphocyte counts were not associated with mortality in adjusted or unadjusted analysis. (Table 5)

In subgroup analysis, we observed substantial variation in the association between laboratory values and mortality by age, sex, and comorbidity status. Elevated white blood cell counts and c-reactive protein levels were risk factors for mortality in those aged 18–64 years (WBC >10.8 $10^3$/μL; aRR 1.75, 95% CI 1.00–3.08: CRP >200 mg/L; aRR 1.83, 95% CI 1.01–3.31) but were not associated with mortality among those 65 years or older (WBC aRR 0.70, 95% CI 0.36–1.35: CRP aRR 0.57, 95% CI 0.26–1.24). Elevated ferritin was a mortality risk factor for females (>1000 mg/dL; aRR 3.60, 95% CI 1.90–6.83) but not males (aRR 1.32, 95% CI 0.77–2.24). Elevated troponin was a risk factor for mortality for males (>0.05 ng/mL; aRR 1.95, 95% CI 1.23–3.09) but not females (aRR 0.81, 95% CI 0.31–2.15). Elevated troponin was also a risk factor among those with no comorbid conditions (aRR 4.25, 95% CI 1.94–9.31) while it was

**Table 3. Hospital outcomes stratified by respiratory support and renal injury for patient with COVID-19 illness requiring supplemental oxygen (n = 270).**

| | Totals | Mortality | p-value[1] | Remain in ICU at HD 30 | Remain on floor at HD 30 | Discharge by HD 30 |
|---|---|---|---|---|---|---|
| **Totals** | **270** | **78 (28.9%)** | | **22 (8.1%)** | **33 (12.2%)** | **137 (50.7%)** |
| **Never ICU. n (%)** | **135** | **6 (4.4%)** | | - | **10 (7.4%)** | **119 (88.2%)** |
| Respiratory Support | | | | | | |
| Nasal Cannula | 91 | 2 (2.2%) | Ref | - | 4 (4.4%) | 85 (93.4%) |
| NIV or NRB/HFNC | 44 | 4 (9.1%) | 0.088 | - | 6 (13.6%) | 34 (77.3%) |
| Renal Injury | | | | | | |
| ESRD | 8 | 0 (0%) | - | - | 3 (37.5%) | 5 (62.5%) |
| No AKI | 103 | 3 (2.9%) | Ref | - | 4 (3.9%) | 96 (93.2%) |
| AKI without RRT | 23 | 3 (13.0%) | 0.074 | - | 2 (8.7%) | 18 (78.2%) |
| AKI with or without RRT[2] | 24 | 3 (12.5%) | 0.081 | - | 3 (12.5%) | 18 (75.0%) |
| **ICU, n (%)** | **135** | **72 (53.3%)** | | **22 (16.3%)** | **23 (17.0%)** | **18 (13.3%)** |
| Respiratory Support | | | | | | |
| NIV or NRB | 14 | 5 (35.7%) | Ref | 0 (0%) | 2 (14.3%) | 7 (50.0%) |
| Mechanical Ventilation | 121 | 67 (55.3%) | 0.257 | 21 (17.4%) | 21 (17.4%) | 12 (9.9%) |
| Renal Injury | | | | | | |
| ESRD | 5 | 4 (80.0%) | - | 0 (0%) | 0 (0%) | 1 (20.0%) |
| No AKI | 31 | 9 (29.0%) | Ref | 5 (16.1%) | 5 (16.1%) | 12 (38.7%) |
| AKI without RRT | 44 | 23 (52.3%) | 0.059 | 7 (15.9%) | 10 (22.7%) | 4 (9.1%) |
| AKI with RRT | 55 | 36 (65.5%) | 0.002 | 10 (18.2%) | 8 (14.5%) | 1 (1.8%) |
| AKI with or without RRT | 99 | 59 (59.6%) | 0.004 | 17 (17.2%) | 18 (18.2%) | 5 (5.0%) |

Data are reported as count and proportion for categorical variables.

[1]P-values for comparison of mortality, stratified by respiratory support and renal injury categories, are calculated by Fisher's exact test. Reference group (Ref) listed. Patients with ESRD are excluded from analysis of renal injury on mortality.

[2]Renal injury groups combined as only one patient in this group had AKI requiring RRT.

Abbreviations: HD Hospital Day: AKI Acute Kidney Injury: RRT Renal Replacement Therapy: NIV Non-invasive Ventilation: NRB Non-rebreather: HFNC High Flow Nasal Cannula: ESRD End-Stage Renal Disease.

weakly and insignificantly associated with mortality among those with at least one comorbidity (aRR 1.43, 95% CI 0.91–2.25).

## Discussion

We describe clinical presentation and outcomes of patients admitted with COVID-19 to a large, safety-net hospital in NYC. In this exploratory analysis, we assessed differential characteristics and outcomes by severity of illness primarily by location of care (ICU vs non-ICU), with a focus on patients requiring supplemental oxygen. Our main findings include identification of dementia as well as obesity, age, and male sex as strong mortality risk factors, a finding of low mortality among patients with severe illness managed outside the ICU, and the clinical impact of renal injury for patients both in and outside the ICU.

Mortality among patients requiring supplemental oxygen, excluding those with mild disease not requiring supplemental oxygen therapy, was 28.9%, with the majority of deaths occurring in the ICU and among those receiving mechanical ventilation. Mortality rates in the ICU and with mechanical ventilation vary widely in previous reports (14.6%-88.1%) [4, 10, 12, 17]. The 30-day mortality rate in the BHC ICU (53.3%) was higher than some other NYC reports. This may have been influenced by BHC's status as a referral center for critically ill patients, including those requiring subspecialty care, for the NYCH+H system. Another potentially mediating factor is duration of symptoms prior to presentation. The median duration of

**Table 4. Unadjusted and adjusted Risk Ratios (RR) and 95% Confidence Intervals (CI) for associations between baseline socio-demographic and baseline comorbidity[1] and COVID-19 mortality (n = 270 patients treated with supplemental oxygen).**

| Baseline Characteristics | N (%) Mortality | RR (95% CI) | Adjusted RR[2] (95% CI) |
|---|---|---|---|
| Age | | | |
| 18–64 years (n = 185) | 41 (22.2) | Ref | Ref |
| 65 years and older (n = 85) | 37 (43.5) | 1.96 (1.37–2.82)* | 1.97 (1.31–2.95)* |
| Sex | | | |
| Female (n = 88) | 22 (25.0) | Ref | Ref |
| Male (n = 182) | 56 (30.8) | 1.23 (0.81–1.88) | 1.32 (1.04–1.70)* |
| Race | | | |
| White (n = 25) | 7 (28.0) | Ref | Ref |
| Black/Latinx (n = 80) | 24 (30.0) | 1.07 (0.53–2.18) | 0.98 (0.50–1.93) |
| Other, Asian, American Indian/Alaska Native (n = 146) | 43 (29.5) | 1.05 (0.53–2.07) | 1.27 (0.65–2.48) |
| Not Defined (n = 19) | 4 (21.1) | 0.75 (0.26–2.20) | 1.21 (0.46–3.19) |
| Cardiovascular Comorbidity[3] | | | |
| No (n = 131) | 32 (24.4) | Ref | Ref |
| Yes (n = 139) | 46 (33.1) | 1.35 (0.92–1.99) | 1.12 (0.75–1.68) |
| Pulmonary Comorbidity[4] | | | |
| No (n = 234) | 68 (29.1) | Ref | Ref |
| Yes (n = 36) | 10 (27.8) | 0.96 (0.54–1.68) | 0.86 (0.51–1.45) |
| Renal Comorbidity[5] | | | |
| No (n = 238) | 67 (28.2) | Ref | Ref |
| Yes (n = 32) | 11 (34.4) | 1.22 (0.73–2.05) | 1.27 (0.90–1.78) |
| Type 2 Diabetes | | | |
| No (n = 181) | 52 (28.7) | Ref | Ref |
| Yes (n = 89) | 26 (29.2) | 1.02 (0.68–1.51) | 0.86 (0.62–1.21) |
| Immunosuppression | | | |
| No (n = 259) | 76 (29.3) | Ref | Ref |
| Yes (n = 11) | 2 (18.2) | 0.62 (0.17–2.20) | 0.63 (0.17–2.36) |
| HIV | | | |
| No (n = 265) | 76 (28.7) | Ref | Ref |
| Yes (n = 5) | 2 (40.0) | 1.39 (0.47–4.15) | 1.46 (0.49–4.31) |
| Malignancy | | | |
| No (n = 263) | 77 (29.3) | Ref | Ref |
| Yes (n = 7) | 1 (14.3) | 0.49 (0.08–3.02) | 0.57 (0.09–3.57) |
| Dementia | | | |
| No (n = 263) | 71 (27.2) | Ref | Ref |
| Yes (n = 9) | 7 (77.8) | 2.86 (1.91–4.27)* | 2.11 (1.50–2.96)* |
| Obesity (Body Mass Index ≥30) | | | |
| No (n = 152) | 41 (27.0) | Ref | Ref |
| Yes (n = 109) | 35 (32.1) | 1.19 (0.82–1.74) | 1.37 (1.07–1.74)* |

*$p < 0.05$.

[1] A total of 211 of 270 patients had at least one comorbidity upon hospitalization including cardiovascular, pulmonary, or renal comorbidity, diabetes, immunosuppression, HIV, malignancy, dementia, obesity (Body Mass Index ≥30). Presentation with any comorbidity was associated with 1.56 times the mortality risk (95%CI:0.87–2.82).

[2] All socio-demographic and comorbidity factors were included in the fully-adjusted model.

[3] Hypertension, Heart Failure, Stroke or Transient Ischemic Attack, Coronary Artery Disease.

[4] Asthma, Chronic Obstructive Pulmonary Disease (COPD), Obstructive Sleep Apnea, Interstitial Lung Disease.

[5] Chronic Kidney Disease (CKD) or End-stage Renal Disease (ESRD).

**Table 5. Unadjusted and adjusted Risk Ratios (RR) and 95% Confidence Intervals (CI) for associations between selected laboratory indicators and COVID-19 mortality (n = 270 patients treated with supplemental oxygen).**

| Laboratory Indicators | N (%) Mortality | RR (95% CI) | Adjusted RR[1] (95% CI) |
|---|---|---|---|
| White Blood Cell >10.8 $10^3$/μL | | | |
| No (n = 209) | 59 (28.2) | Ref | Ref |
| Yes (n = 61) | 19 (31.2) | 1.10 (0.72–1.70) | 1.09 (0.72–1.65) |
| | | *Stratified by Age:*[2] | *Stratified by Age:*[2] |
| | | *18–64 yrs: 1.60 (0.90–2.83)* | *18–64 yrs: 1.75 (1.00–3.08)* |
| | | *65+ yrs: 0.63 (0.32–1.23)* | *65+ yrs: 0.70 (0.36–1.35)* |
| Absolute Neutrophil Count (ANC) >7.6 $10^3$/μL | | | |
| No (n = 189) | 47 (26.1) | Ref | Ref |
| Yes (n = 89) | 31 (34.8) | 1.33 (0.92–1.94) | 1.28 (0.89–1.86) |
| Absolute Lymphocyte Count (ALC) <0.8 $10^3$/μL | | | |
| No (n = 158) | 41 (26.0) | Ref | Ref |
| Yes (n = 111) | 37 (33.3) | 1.28 (0.89–1.86) | 1.11 (0.76–1.61) |
| Aspartate Transaminase (AST) >40 U/L | | | |
| No (n = 80) | 16 (20.0) | Ref | Ref |
| Yes (n = 185) | 60 (32.4) | 1.62 (1.00–2.63) | 1.84 (1.11–3.04) |
| C-reactive protein (CRP) >200 mg/L | | | |
| No (n = 168) | 45 (26.8) | Ref | Ref |
| Yes (n = 62) | 19 (30.7) | 1.14 (0.73–1.79) | 1.03 (0.66–1.59) |
| | | *Stratified by Age:*[2] | *Stratified by Age:*[2] |
| | | *18–64 yrs: 1.79 (0.99–3.23)* | *18–64 yrs: 1.83 (1.01–3.31)\** |
| | | *65+ yrs: 0.62 (0.28–1.37)* | *65+ yrs: 0.57 (0.26–1.24)* |
| Ferritin >1000 ng/mL | | | |
| No (n = 107) | 24 (22.4) | Ref | Ref |
| Yes (n = 80) | 27 (33.8) | 1.50 (0.94–2.40) | 1.66 (1.05–2.63) |
| | | | *Stratified by Sex:*[2] |
| | | | *Female: 3.60 (1.90–6.83)\** |
| | | | *Male: 1.32 (0.77–2.24)* |
| Lactate dehydrogenase (LDH) >500 IU/mL | | | |
| No (n = 106) | 19 (17.9) | Ref | Ref |
| Yes (n = 117) | 44 (37.6) | 2.10 (1.31–3.36)* | 2.12 (1.32–3.41) |
| Troponin >0.05 ng/mL | | | |
| No (n = 146) | 39 (26.7) | Ref | Ref |
| Yes (n = 43) | 19 (44.2) | 1.65 (1.08–2.54)* | 1.57 (1.03–2.41) |
| | | *Stratified by Sex:*[2] | *Stratified by Sex:*[2] |
| | | *Female: 0.92 (0.35–2.40)* | *Female: 0.81 (0.31–2.15)* |
| | | *Male: 2.09 (1.29–3.36)\** | *Male: 1.95 (1.23–3.09)\** |
| | | | *Stratified by Comorbidity:*[2] |
| | | | *No Comorbidity: 4.25 (1.94–9.31)\** |
| | | *Stratified by Comorbidity:*[2] | *≥1 Comorbidity: 1.43 (0.91–2.25)* |
| | | *No Comorbidity: 5.41 (2.64–11.09)\** | |
| | | *≥1 Comorbidity: 1.41 (0.89–2.25)* | |

*p<0.05.

[1]Adjusted for age (65 years or older versus 18–64 years), sex, race, and presence of at least one comorbidity upon hospitalization (cardiovascular, pulmonary, or renal comorbidity, diabetes, immunosuppression, HIV, malignancy, dementia, obesity).

[2]Interactions tested for age, sex, and presence of any comorbidity. Stratified estimates only included for interactions that were significant at the p<0.15 level.

symptoms in our population was longer (7 days) than other NYC cohorts, although most studies do not report this variable and it is prone to recall and reporter bias [10]. One study from Seattle which showed similar mortality rate (50%) in critically ill patients had similar duration of symptoms prior to presentation [9].

Mortality for those managed outside the ICU was 4.4%, including for those managed for severe hypoxia with escalation to NRB or NIV/HFNC, similar or lower than what was observed in other NYC cohorts [4, 12, 17]. Several interventions allowed for safe management of higher acuity patients outside the ICU, including the following: 1) An increase in continuous oxygen monitoring capabilities, allowing for rapid detection of decompensations and situations that required escalations in care; 2) Expansion of rapid response and code teams ensuring a timely response to these decompensating patients; 3) An increase in nurse staffing ratios on the floor to 4:1 from the standard of 6:1; and 4) Newly developed oxygenation protocols to guide frontline providers in management of these patients.

Risk factors associated with mortality reported in prior studies include advanced age, male sex, obesity, and other comorbidities, specifically cardiovascular conditions such as hypertension [4, 12, 18, 19]. In our population, advanced age, male sex, and obesity were the main factors associated with mortality. Cardiovascular and renal comorbidities were not associated with mortality in an adjusted analysis, perhaps owing to a limited sample size and inability to detect these associations. Alternatively, our findings could suggest the relative importance of age and obesity in a younger population with fewer medical comorbidities. In contrast with other cohorts from NYC, where fewer than 10% of patients had no comorbid conditions, approximately one in five patients in our cohort had no recognized comorbid conditions. However, we cannot assess in our analysis whether the lower comorbidity rate is related to unrecognized comorbid conditions in a vulnerable population without adequate access to primary care services.

Low rates of baseline immunosuppression, HIV, and liver disease precluded adequate assessment of the effect that these comorbid conditions may have on COVID-19 severity and outcomes. Despite a small population of patients with pre-existing dementia, this comorbidity was strongly associated with mortality. This could reflect prior observations of increased mortality related to pneumonia among those with pre-existing dementia [20]. On dedicated review, we noted that do-not-resuscitate orders were in place for 5 of the 7 patients with dementia at the time of death, and therefore a lower likelihood of care escalation with measures such as resuscitation or intubation may have led to higher rates of mortality. However, code status was unavailable for the full cohort, limiting comparative analyses. Additionally, all patients with dementia were over the age of 65, and compared to the full cohort of individuals over 65, their age was on average 8 years greater. This raises the possibility of additional confounding based on more advanced age not accounted for in our adjustment by age category.

Laboratory abnormalities associated with mortality included ferritin and LDH. Additionally, markers of multi-organ involvement such as transaminase and troponin elevation were associated with mortality. Despite a relatively small cohort, subgroup analyses demonstrated interesting associations, including the correlation of ferritin with mortality specifically among female patients and the association of troponin elevation with mortality among male patients. These observations could help tailor laboratory assessment at the time of admission for patients diagnosed with COVID-19, and direct the development of predictive models in larger datasets in the future.

Despite a low prevalence of baseline renal comorbidities, AKI and need for RRT were seen in a high proportion of critically ill patients, consistent with other NYC reports [4, 10]. The expanded need for RRT was unexpected, and has led to a significant strain on dialysis capability across hospital systems [21, 22]. The need for expanded dialysis access should be expected

for hospitals responding to a surge in COVID-19 illness unless current and future therapies employed against COVID-19 can mitigate the rate of renal injury in COVID-19 critical illness.

Rates of RRT in critically ill populations in NYC are similar across different hospitals, ranging from approximately 30–40% in most reports [4, 10, 12]. Many hypotheses for high rates of AKI and RRT in critical illness in COVID-19 have been proposed. As suggested by other researchers, the fluid-restrictive strategy in management of ARDS could cause unintended harm by worsening AKI, which may explain similar rates of RRT in ICU cohorts [12]. Additionally, pathologic studies have identified that viral entry and replication in renal cells via ACE-2 receptors may point to a direct impact on renal injury, which could be exaggerated during uncontrolled viral replication in critical illness [23, 24].

Regarding treatment of patients with COVID-19 illness, this cohort of patients notably presented during a period when scant evidence was available for guidance of treatment strategies. We observed high rates of hydroxychloroquine usage, although subsequent observational and randomized trials have failed to show significant benefit in treatment of COVID-19 illness [25–27]. Steroids were employed frequently for critically ill patients in our study but rarely used for patients requiring supplemental oxygen outside the ICU. Steroid therapy has subsequently shown promise to reduce mortality in patients receiving mechanical ventilation and supplemental oxygen compared to usual care [28]. We anticipate that many of the patients requiring supplemental oxygen but not requiring ICU admission would be treated with steroids under current treatment guidelines. Fewer numbers of critically ill patients received tocilizumab, likely owing to limited familiarity with this medication and a higher perceived risk profile. The role of targeted immune suppressants such as tocilizumab remains to be elucidated; while early observational data suggested a benefit, subsequent randomized trials have not demonstrated benefit [29]. Remdesivir, which had limited availability during this study period through a randomized trial, has subsequently shown promise in reducing time to recovery although did not demonstrate a significant mortality reduction in a randomized trial [30]. Finally, we observed high rates of concurrent antibiotic usage for management of patients both in and outside the ICU despite low rates of documented co-infection [31]. Widespread usage of broad-spectrum antimicrobials without clear benefit outside of documented bacterial infection raise the concern about accelerating antimicrobial resistance in vulnerable populations in the face of the pandemic.

Strengths of this study include that data was gathered by manual chart review. This assures that information not well recorded in extractable areas in the EMR (e.g. symptoms and symptom duration) were accurately gathered. Among adult patients, only those incarcerated were excluded as a vulnerable population; therefore, this study is representative of the outbreak at our single institution for the first month of the COVID-19 outbreak in NYC.

This study has several limitations. As a retrospective chart review, data is restricted to what is documented in the EMR, with the possibility of patient-recall error or documentation error. The retrospective analysis is also restricted to the clinical data gathered, leading to missing data points. Detailed race and ethnicity were not recorded for many patients in this study, owing likely to limitations with demographic data entry during surge in clinical volume, limiting our ability to robustly analyze presentation and outcomes by race and ethnicity.

## Conclusion

We describe a cohort of patients managed at a public hospital in NYC during the first month of the COVID-19 pandemic. BHC, along with the entire NYCH+H system, made significant structural changes to care for the vulnerable populations seen at our institution to provide a high level of care and mitigate to the extent possible the morbidity and mortality associated

with COVID-19. Our results shed light on important clinical characteristics and outcomes in public hospitals. Future work is needed to continue to understand the pathophysiology of COVID-19 and develop best practices for the management of hospitalized patients with COVID-19.

## Supporting information

**S1 Table. Socio-demographics, medical history, admission laboratory markers, treatment, and hospital outcomes for patients not requiring supplemental oxygen (n = 65).**
(DOCX)

**S2 Table. Smoking, exposures, and presenting vital signs for patients with COVID-19 illness requiring supplemental oxygen (n = 270).**
(DOCX)

## Author Contributions

**Conceptualization:** Thomas D. Filardo, Hayley Galitzer, Savannah Karmen-Tuohy, Megan Coffee, Verity E. Schaye, Benjamin J. Eckhardt, Gabriel M. Cohen.

**Data curation:** Thomas D. Filardo, Maria R. Khan, Noa Krawczyk, Benjamin J. Eckhardt, Gabriel M. Cohen.

**Formal analysis:** Thomas D. Filardo, Maria R. Khan, Noa Krawczyk, Megan Coffee, Benjamin J. Eckhardt, Gabriel M. Cohen.

**Investigation:** Thomas D. Filardo, Hayley Galitzer, Savannah Karmen-Tuohy.

**Supervision:** Thomas D. Filardo, Benjamin J. Eckhardt, Gabriel M. Cohen.

**Validation:** Thomas D. Filardo.

**Visualization:** Maria R. Khan, Noa Krawczyk.

**Writing – original draft:** Thomas D. Filardo.

**Writing – review & editing:** Thomas D. Filardo, Maria R. Khan, Noa Krawczyk, Hayley Galitzer, Savannah Karmen-Tuohy, Megan Coffee, Verity E. Schaye, Benjamin J. Eckhardt, Gabriel M. Cohen.

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
