## [Decision Letter · Decision Letter 0]

1 Sep 2020

PONE-D-20-23829

Comorbidity and Clinical Factors Associated with COVID-19 Critical Illness and Mortality at a Large Public Hospital in New York City in the Early Phase of the Pandemic (March-April 2020)

PLOS ONE

Dear Dr. Thomas Daniel Filardo,

Thank you for submitting your manuscript to PLOS ONE. After careful consideration, we feel that it has merit but does not fully meet PLOS ONE’s publication criteria as it currently stands. Therefore, we invite you to submit a revised version of the manuscript that addresses the points raised during the review process.

ACADEMIC EDITOR: The reviewers have raised a number of points which we believe major modifications are necessary to improve the manuscript, taking into account the reviewers' remarks. Please consider and address each of the comments raised by the reviewers before resubmitting the manuscript. This letter should not be construed as implying acceptance, as a revised version will be subject to re-review.

We look forward to receiving your revised manuscript.

Kind regards,

Wisit Cheungpasitporn, MD

Academic Editor

PLOS ONE

Journal Requirements:

2. Please include the date(s) on which you accessed the databases or records to obtain the data used in your study.

3. For studies involving humans categorized by race/ethnicity, age, disease/disabilities, religion, sex/gender, sexual orientation, or other socially constructed groupings, authors should:

1) Explicitly describe their methods of categorizing human populations,

2) Define categories in as much detail as the study protocol allows,

3) Justify their choices of definitions and categories,

4) Explain whether (and if so, how) they controlled for confounding variables such as socioeconomic status, nutrition, environmental exposures, or similar factors in their analysis, and

5) Update outmoded terms and potentially stigmatizing labels to more current, acceptable terminology.

Examples: “Caucasian” should be changed to “white” or “of [Western] European descent” (as appropriate); “XXX victims” should be changed to “patients with XXX.

Reviewers' comments:

Reviewer's Responses to Questions

**Comments to the Author**

1. Is the manuscript technically sound, and do the data support the conclusions?

Reviewer #1: Yes

Reviewer #2: Yes

Reviewer #3: Partly

Reviewer #4: Partly

Reviewer #5: Yes

2. Has the statistical analysis been performed appropriately and rigorously? 

Reviewer #1: No

Reviewer #2: I Don't Know

Reviewer #3: No

Reviewer #4: I Don't Know

Reviewer #5: Yes

3. Have the authors made all data underlying the findings in their manuscript fully available?

Reviewer #1: Yes

Reviewer #2: Yes

Reviewer #3: Yes

Reviewer #4: No

Reviewer #5: Yes

4. Is the manuscript presented in an intelligible fashion and written in standard English?

Reviewer #1: Yes

Reviewer #2: Yes

Reviewer #3: Yes

Reviewer #4: Yes

Reviewer #5: Yes

5. Review Comments to the Author

Reviewer #1: 1. * Table 1: means for skewed continuous variables (labs) need to be compared using a non-parametric test, such as the Kruskal-Wallis test, not a between-group t-test, which assumes normal distributions.

2. * The authors state “... altered mental status was more common in patients requiring ICU admission (9.6% vs 5.2%, p = 0.16) ..“ When a difference is not significant, the proper way to report it would be something like “ ...altered mental status did not differ significantly between those who did vs did not require ICU admission (9.6% vs 5.2%, p = 0.16). When the p value is > .05 and < 0.10, it may be acceptable to report a difference ‘with a trend level of significance’.

3. It is confusing to have “Supplemental Oxygen” in the table headings in Tables 1 and 2. I would suggesting removing this so it is clear the columns refer to ICU and no-ICU.

4. * Interactions are mentioned in the Method section and are shown in Table 5, but are never discussed in the Results or Discussion.

Reviewer #2: Persons with dementia seem more sensitive to certain types of immune challenge and show higher mortality. Are you able to do any further analysis of the dementia group to suggest reasons for their susceptibility to death from Covid-19?

Given the rapid developments in Covid-19 treatment could the authors make additional comments on the treatment regimes or recent relevant studies.

Reviewer #3: The manuscript entitled ‘Comorbidity and Clinical Factors Associated with COVID-19 Critical Illness and Mortality at a Large Public Hospital in New York City in the Early Phase of the Pandemic (March-April 2020)’ with the aim to assess demographics, comorbid conditions, and clinical factors associated with critical illness and mortality among patients diagnosed with COVID-19 at a public hospital in New York City (NYC) during the first month of the COVID-19 outbreak.

This is an interesting study but the manuscript can be further improved in terms of presentation.

Comments

Abstracts

n= to be inserted for the number of subjects i.e n=127

Methods

Data

For the ‘team of students’, who are the students to be clearly stated.

Data analysis

All statistical tests used in the results section to be clearly stated in statistical analyses section. More information to be provided after this statement ‘testing effect modification by sex, age and comorbidity at the 0.15 level’. The acceptance level of statistical significant to be stated.

Results

Table 1, n= to be employed to indicate total sample. X^2 to be replaced with the actual ‘chi square' symbol.

n(%), N(%) to be placed in the first row. Individual symbol % for individual figures to be omitted. Statistical tests to be clearly denoted in table footnote.

Table 1, under Admission Laboratory Tests, n to be placed after the variable name and not after median (IQR) values (Likewise with Supplemental Table 1). The categorical variables and the continuous variable to be clearly separated with space and indicated with their respective p value. Row alignment for CRP to be improved. Symbol >= to be replaced with symbol ≥ (likewise in Supplemental Table 1). In Supplemental Table 2 <= to be replaced with ≤)

Table 2 & 3 n(%) N(%) to be placed in the first row. Symbol % for individual figure to be omitted. Likewise with Supplemental Table 1 & Supplemental Table 2.

Table 2, NRB to be placed after Oxygen-Non-Rebreather.

Treatment

The sentence ‘The majority of patients received hydroxychloroquine (82.4%)’ to be revised and match the variable name in Table 2. HQ and any HCQ to be clearly defined/denoted in table footnote.

Hospital outcomes

Table 3, footnote X^2 to be replaced with symbol ‘chi square' .

Table 3 to be cited after ‘ In critically ill patients …. P=0.045)

Clinical Presentation

For ‘Mean NEWS score was higher for patients admitted to the ICU (7.0 vs 5.9, p<0.01)’, median was reported in Table 1.

Table 4 & 5, small n to be used to indicate individual sample size. N for overall total.

Table 5, the description for superscript 2,3,4 in the footnote requires fine tuning.

Symbol colon : for the sub title to be omitted.

Some references were not conformed to the journal format.

Tables formatting to follow journal format.

Reviewer #4: Review for PLoS ONE

Aug 17, 2020

Title: Comorbidity and Clinical Factors Associated with COVID-19 Critical Illness and Mortality at a Large Public Hospital in New York City in the Early Phase of the Pandemic (March-April 2020)

This paper presents a study to describe patients admitted to NYC Health + Hospitals / Bellevue Hospital Center, in NYC, with COVID-19 in the month following the hospital’s first diagnosed case. Authors focused on comorbidities, clinical characteristics, and outcomes for patients admitted to ICU and non-ICU settings. However, there are questions that limit my enthusiasm of the paper, as outlined below.

1. Data Analysis:

a. Authors considered IQR, median, frequency, t-test and chi-squared tests, while they weren’t introduced at data analysis section. Please explain the descriptive statistics (for both continuous and categorical variables) along with the association tests that were applied across manuscript including t-test and chi-squared test. (Numbers 1 and 7 of caption of Table 1 need to be transferred to the data analysis with more details.)

b. Why authors considered parametric t-test compared with non-parametric methods? Is it based on known or unknown variance? Please justify this part since for the parametric t-test, the normality distribution needs to be assessed before applying t-test.

c. As an example, Table shows the association between race and requiring supplemental Oxygen. How authors consider the chi-squared test for such analysis with few observations or even zero values? Please adjust such analysis by applying an appropriate statistical analysis (such as Fisher’s exact test) and add to the data analysis section as well.

d. What is the statistical significant threshold for the statistical analyses across manuscript? Please add to the manuscript.

e. Statistically, the logistic regression model is fitted not estimated. But we can consider the logistic regression model to estimate the OR and/or RR. In addition, bivariate analyses or bivariable analyses? Please clarify this part at data analysis section.

2. Results:

a. For NEWS score, authors considered mean or median? Page 13, authors mentioned mean, so please be consistent across manuscript to report descriptive statistics.

b. Table 1, not easy to follow. For example for symptom duration or NEWS, it’s supposed to have (median, IQR), but what are those numbers after the IQR values? If they represent the total observations then please modify the table and re-do the Table in better way.

c. Table 1, for the CRP (admission lab test), please report all required values.

d. Please define “other” group in race as a caption or across the manuscript.

e. Why authors report cirrhosis (medical history), while it doesn’t show any observation/data?

f. Table 3: not easy to follow the association analysis for outcomes. Please add the P values (in the caption) to the Table and make it easier to follow. In addition, apply the previous comments regarding the chi-square test.

g. For the sum of the percentage of contingency table, it should be always 100%. So please modify the Table and exclude those two patients. In addition, the sum of the supplemental Oxygen, ICU is 135 not 136? Please modify Table 3 not easy to follow.

h. For the adjusted logistic model, how the authors decide which variables or covariates need to be included in the model to adjust the fitted model?

Reviewer #5: Thank you for the opportunity to review the well reported manuscript.

In general, the data analyses were appropriately performed and reported. Minor amendments with a recheck on percentage calculations throughout all tables were needed for a quality presentation. Detailed comments that focused mainly on the data analysis can be found from the attachment.

6. PLOS authors have the option to publish the peer review history of their article (what does this mean?). If published, this will include your full peer review and any attached files.

Reviewer #1: No

Reviewer #2: No

Reviewer #3: No

Reviewer #4: No

Reviewer #5: No

---

## [Author Response · Author response to Decision Letter 0]

14 Oct 2020

We have updated file naming conventions based on reviewing author guidelines.

Regarding classification of individuals as part of this study, we have updated out methods section to include our categorization of race and ethnicity. This was also suggested by reviewers, and is included in our response to reviewers. We hope that this provides clarification of our study methodology.

Finally, we have included in our Methods section the dates of access of the electronic medical record as requested (Page 3).

---

## [Decision Letter · Decision Letter 1]

10 Nov 2020

Comorbidity and Clinical Factors Associated with COVID-19 Critical Illness and Mortality at a Large Public Hospital in New York City in the Early Phase of the Pandemic (March-April 2020)

PONE-D-20-23829R1

Dear Dr. Filardo,

We’re pleased to inform you that your manuscript has been judged scientifically suitable for publication and will be formally accepted for publication once it meets all outstanding technical requirements.

Kind regards,

Wisit Cheungpasitporn, MD

Academic Editor

PLOS ONE

Additional Editor Comments (optional):

I reviewed the revised manuscript and the response to reviewers' comments. Revised Manuscript is well written. All comments have been addressed and thus accepted for publication

Reviewers' comments:

Reviewer's Responses to Questions

**Comments to the Author**

1. If the authors have adequately addressed your comments raised in a previous round of review and you feel that this manuscript is now acceptable for publication, you may indicate that here to bypass the “Comments to the Author” section, enter your conflict of interest statement in the “Confidential to Editor” section, and submit your "Accept" recommendation.

Reviewer #2: All comments have been addressed

Reviewer #3: All comments have been addressed

Reviewer #4: All comments have been addressed

Reviewer #5: All comments have been addressed

2. Is the manuscript technically sound, and do the data support the conclusions?

Reviewer #2: Yes

Reviewer #3: Partly

Reviewer #4: Yes

Reviewer #5: Yes

3. Has the statistical analysis been performed appropriately and rigorously? 

Reviewer #2: I Don't Know

Reviewer #3: No

Reviewer #4: Yes

Reviewer #5: Yes

4. Have the authors made all data underlying the findings in their manuscript fully available?

Reviewer #2: Yes

Reviewer #3: Yes

Reviewer #4: Yes

Reviewer #5: Yes

5. Is the manuscript presented in an intelligible fashion and written in standard English?

Reviewer #2: Yes

Reviewer #3: Yes

Reviewer #4: Yes

Reviewer #5: Yes

6. Review Comments to the Author

Reviewer #2: Thank you for your hard work and the thought given in the revision. Hopefully this will contribute to the wider body of research on Covid-19

Reviewer #3: The authors have put in great effort to address the comments

For Fisher's Exact Test, 1 or 2 tailed test to be stated.

Reviewer #4: All comments have been addressed. Thank you!

Minor comment, please make sure to use right name for Fisher's exact test across the manuscript (Please check caption Table 3). Please note that for enough sample, it is appropriate to consider the Chi-squared test. However for small sample size, the Fisher's exact test is the appropriate one.

Reviewer #5: All comments have been fully addressed and hence the manuscript has been accepted by me for publication.

7. PLOS authors have the option to publish the peer review history of their article (what does this mean?). If published, this will include your full peer review and any attached files.

Reviewer #2: **Yes: **Rodney P Jones

Reviewer #3: No

Reviewer #4: No

Reviewer #5: No

---

## [Editor Report · Acceptance letter]

13 Nov 2020

PONE-D-20-23829R1 

Comorbidity and Clinical Factors Associated with COVID-19 Critical Illness and Mortality at a Large Public Hospital in New York City in the Early Phase of the Pandemic (March-April 2020) 

Dear Dr. Filardo:

I'm pleased to inform you that your manuscript has been deemed suitable for publication in PLOS ONE. Congratulations! Your manuscript is now with our production department. 

Kind regards, 

on behalf of

Dr. Wisit Cheungpasitporn 

Academic Editor

PLOS ONE